# Oral Abscess Caused by *Chryseobacterium indologenes* in Ball Python (*Python regius*); A Case Report

**DOI:** 10.3390/antibiotics10060686

**Published:** 2021-06-08

**Authors:** Iradj Ashrafi Tamai, Babak Pakbin, Zahra Ziafati Kafi, Wolfram Manuel Brück

**Affiliations:** 1Department of Microbiology and Immunology, Faculty of Veterinary Medicine, University of Tehran, Tehran 1417466191, Iran; iashrafi@ut.ac.ir (I.A.T.); Zahra.ziafati@ut.ac.ir (Z.Z.K.); 2Institute for Life Technologies, University of Applied Sciences Western Switzerland Valais-Wallis, 1950 Sion 2, Switzerland

**Keywords:** *Chryseobacterium indologenes*, oral abscess, ball python

## Abstract

*Chryseobacterium indologenes* is an opportunistic pathogen isolated from human infections and, rarely, from some aquatic animals. A 3-year-old male ball python (*Python regius*) was admitted to the veterinary clinic by a pet owner because of acute respiratory and swallowing failure. During physical examinations, oral secretions and abscesses were observed in the mouth cavity and throat of the animal. After microbiological analysis including isolation, identification, and 16s rRNA sequencing, *C. indologenes* was detected as the main cause of the oral abscess in this case. Phylogenetic relatedness analysis showed a close relationship between this isolate and other strains isolated from human infections. Antimicrobial susceptibility testing revealed that the isolate was multi-drug resistant. However, it was very sensitive to minocycline, ceftazidime, and tetracycline. The patient was treated by antibiotic therapy and completely recovered after two weeks. To the best of our knowledge, this is the first incidence of *C. indologenes* in an oral abscess in a ball python. As a result we would consider this organism as an opportunistic animal pathogen with zoonotic potentiality.

## 1. Introduction

*Chryseobacterium* species are gram-negative bacteria and have sporadically and ubiquitously been found in the environment. These bacteria have often been detected in water, soil, waste, food sources, domestic animals, and aquatic environments. It was indicated that *Chryseobacterium* species are emerging multi-drug resistant opportunistic nosocomial pathogens and causes of different serious infections in neonates, pregnant women and immunocompromised patients exposed to medical devices and environmental contaminants [1]. Recent research indicated *Chryseobacteria* as the cause of some infectious disease outbreaks in fish and aquatic animals. *C. balustinum*, *C. scophtalmum*, *C. joostei*, *C. piscicola* and *C. arothri* have been isolated from fish species such as salmon and rainbow trout [2].

*C. indologenes* have also been reported as the main infectious agent of fatal diseases in newborns and other *Chryseobacterium* infections in human [3]. However, *C. indologenes* are occasionally isolated in some aquatic species such as yellow perch, Mediterranean limpets, mussels, and purple sea urchins. The isolates were mainly recovered from livers, kidneys, gills, and skin lesions. In these previous studies, all isolates were multi-resistant to a wide range of antibiotics [4,5]. Investigation of genetic relatedness among the isolates showed the evolutionary patterns of the pathogens [6]. Isolation of *Chryseobacterium* species such as *C. indologenes* from terrestrial sources has not been reported yet. In the present study, we investigated the first case of an infected ball python (Pythonidae) with an oral abscess, which, to our knowledge is the first case of *Chryseobacterium* infection in terrestrial animals, caused by multidrug resistant *C. indologenes*.

## 2. Case Presentation

A three-year-old, 22 kg, male ball python (*Python regius*) was submitted by a pet owner to the veterinary clinic at the Faculty of Veterinary Medicine, University of Tehran, Tehran, Iran due to respiratory failure, difficulty swallowing, oral secretions, bad breath, aggression and anorexia. The snake had been fed on small prey, including rodents and birds, and kept in a terrarium. The temperature and relative humidity in the terrarium were 35 °C and 80% RH. Clinical assessments of the snake were performed by the veterinarian of the center. Respiratory problem, bad breath, oral secretions, and several acute abscesses were observed in the mouth and throat of the snake by the veterinarian. Abscess samples were collected for microbiological analysis and isolation of the infectious agents of oral abscesses presented in the infected snake by oral surgery (Figure 1).

Gram-negative bacilli were isolated from the abscess samples and the identities of the isolates were confirmed by biochemical tests. Antibiotic susceptibility of the isolates was evaluated using Kirby–Bauer disk diffusion as described by the Clinical and Laboratory Standards Institute [7]. Biochemical analysis indicated that the isolate was beta-hemolytic; non-motile; indole positive; citrate negative; both methyl red and Voges–Proskauer’s negative; glucose was fermented with gas production; it was negative for urea and unable to grow on MacConkey agar at 42 °C. Antibiotic susceptibility testing showed that the isolate was sensitive to minocycline, ceftazidime, and tetracycline; intermediately sensitive to amikacin, ceftriaxone, ciprofloxacin, enrofloxacin, and levofloxacin; and resistant to imipenem, meropenem, penicillin, gentamycin, erythromycin, and azithromycin antibiotics. 16s rRNA gene sequencing was performed for definite species identification and construction of the phylogenetic tree of the isolate. The 16s rRNA gene of the isolate was amplified by conventional polymerase chain reaction (PCR) using the universal primers pA (5′-AGA GTT TGA TCC TGG CTC AG-3′) and pH (5′-AAG GAG GTG ATC CAG CCG CA-3′) [8]. PCR products were extracted, purified, and sequenced by GATC company (Cologne, Germany). Sequencing of the 16s rRNA gene (NCBI, GenBank accession number: MT276587) indicated that the isolate was *Chryseobacterium indologenes*. CLC software (CLC Genomic WorkBench, Qiagen, Hilden, Germany) was used to generate the cladogram by the neighbor-joining clustering algorithm, bootstrapped 1000 times (using the 16s rRNA gene sequence of the isolate and the NCBI GenBank database). The phylogenetic relationships among the *C. indologenes* isolated in the present study and the strains which were previously isolated in other studies are illustrated in Figure 2. Comparative 16s rRNA gene sequence analysis revealed high phylogenetic relatedness and sequence similarity between the 16s rRNA gene sequence of the strain isolated from oral abscess in the python in the present study and those of the *C. indologenes* strains isolated from human infections in previous studies.

*The C. indologenes* isolated from the oral abscesses in the python was sensitive to minocycline, ceftazidime, and tetracycline. The patient was treated with antibiotic therapy using tetracycline in a dose of 20 mg·kg^−1^ every 48 h for 14 days. During the treatment, the patient was kept in a terrarium and was fed on small prey. All equipment was disinfected with 10% formalin after each experimental session. After 2 weeks the patient was examined physically and microbiologically. Neither respiratory nor swallowing failure associated with the disease was observed in the patient. All oral and throat abscesses were treated and removed, leading to a complete recovery. Regarding the clinical results of the present study, minocycline, ceftazidime, and tetracycline antibiotics are recommended for use as the main treatment of oral abscesses caused by *C. indologenes* in pythons.

## 3. Discussion

Most recently, *C.* Indologenes has been identified in a 52-year-old patient with end stage renal disease (ESRD) [9]. Multi-drug resistant nosocomial and opportunistic pathogens have been regarded as a major challenge among infectious diseases around the globe and have the potential to spread quickly [10]. *C. indologenes* has previously been isolated from some aquatic animal diseases, including animals such as salmon and rainbow trout [5]. There have not been any studies that reported *Chryseobacterium* species as the cause of oral or throat abscesses in human and animals as reported here. As part of the commercial animal trade, pythons are kept in zoos, animal parks, or as domestic pets and should be kept in a terrarium under appropriate temperature and relative humidity conditions [11]. In the case of the present study, the pet owner had kept the python in a suitable terrarium, fed on small prey and water.

This is the first case report presenting oral infection with *Chryseobacterium* in a ball python. Lee and Kim (2011) reported methicillin resistant staphylococcus aureus (MRSA) as the main cause of sub-spectacular abscess in a snake with respiratory failure symptoms. MRSA is a well-known zoonotic gram-positive pathogen around the world which has been reported as the major cause of serious abscess in many animal cases and as a public health concern; consequently, detection and isolation of this pathogen from abscesses is not improbable [12]. Another case study reported by Kurniawan and Govendan (2020) was about a Burmese python with sub-spectacular abscesses. They found oral cavity inflammation and respiratory problems in the patient. However, no microbiological test was performed for the detection of abscess-causing pathogens and the infection was successfully treated by antibiotics including oxytetracycline, meloxicam and enrofloxacin [13]. *C. indologenes* have often been detected in infections with fatal outcomes in neonates. Further, there are several studies about the isolation of this pathogen from immunocompromised patients, the surface of the dwelling devices, as well as hospital vials, tubes, and tap water, confirming its status as a nosocomial pathogen [14]. The close genetic relatedness between the isolate in this report and the isolates from humans reported in other case studies indicates a pathogenic potential of *C. indologenes* isolate in this study for humans. It is also probable that the animal was infected from its prey during feeding [12].

Antimicrobial resistance patterns of the isolate are different from those of the isolates obtained from human infections reported by other researchers. The isolate found here was more sensitive than nosocomial isolates of *C. indologenes* (from human infections) to different antibiotics. Generally, non-nosocomial isolates showed resistance to fewer antibiotics than we have observed in the present report. However, resistance to imipenem and meropenem antibiotics has been observed for both nosocomial and non-nosocomial *C. indologenes* isolates [15].

## 4. Conclusions

It is worth noting that some pets may asymptomatically transmit some pathogens to humans causing lethal emerging infectious diseases. These pets can be considered as an important source of emerging zoonotic diseases, especially when a pathogen leads to infectious diseases such as abscesses or meningitis in both animals and humans [16]. However, in this report, we observed *C. indologenes* as the cause of an oral abscess in a pet python. Due to its differing antimicrobial susceptibility pattern, it may be possible that this isolate does not have the potential to become a human pathogen. It is suggested that for future studies, the antibiotic resistance and virulence factor encoded in the genes of *C. indologenes* isolates from both human and animal infections should be comprehensively investigated to further understand the microorganism’s role as an emerging zoonotic human pathogen.

## Figures and Tables

**Figure 1 antibiotics-10-00686-f001:**
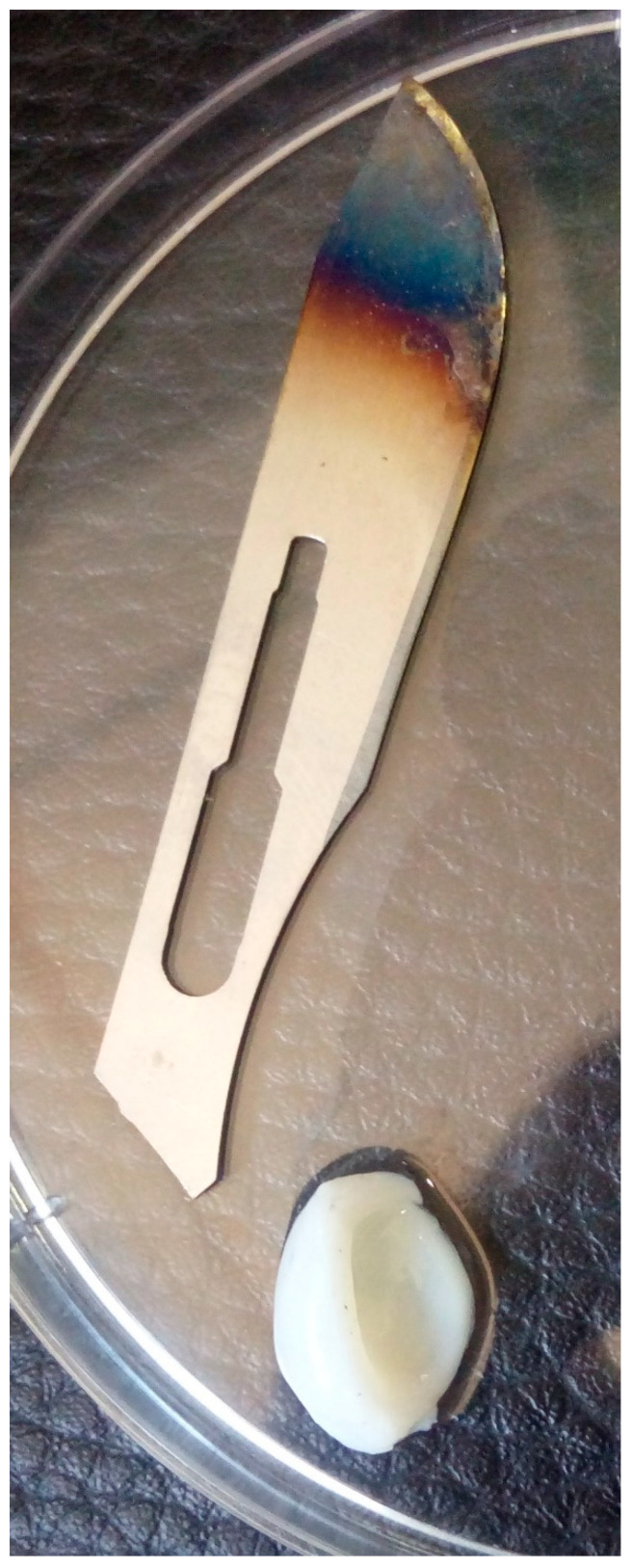
Oral abscess sample separated from the Ball Python mouth.

**Figure 2 antibiotics-10-00686-f002:**
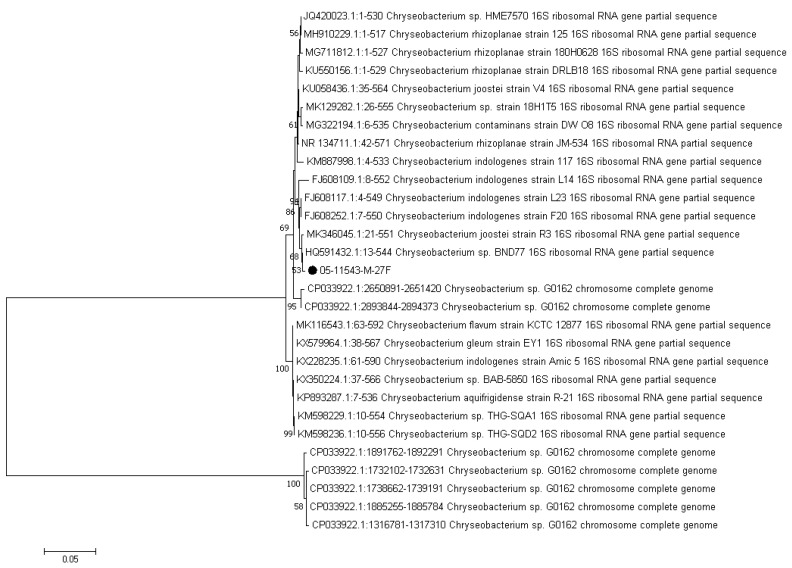
Phylogenetic tree based on 16s rRNA sequencing and neighbor-joining method classification among the strain isolated in this report (NCBI, GenBank accession number: MT276587) and the strains isolated from humans reported by other researchers published in NCBI.

## Data Availability

All data in this study are available from the corresponding authors on a reasonable request.

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
