# Peer review of "Oral Abscess Caused by Chryseobacterium indologenes in Ball Python (Python regius); A Case Report"

_antibiotics, 2021, doi:10.3390/antibiotics10060686_

Round 1

Reviewer 1 Report

This is a very straightforward description of a novel pathogen found in a pet snake and of the successful treatment of the pathogen with a standard antibiotic.  I have only three substantive questions, and only one of those is significant.

  • I am skeptical of the statement on Line 116 that “[t]here are very limited reports and studies about oral or mouth abscess in snakes.” Oral infections are very common in captive snakes, and a number of books concerning veterinary care of reptiles (such as those by Dr. Frederic Frye) have discussed the treatment of those infections.  A simple Google search for “oral abscess snakes” returns a very large number of hits.  That does not mean that this report is not novel, but it does suggest that a more thorough or accurate statement about oral infections in snakes, in general, is warranted.
  •  
  • Of less importance is the mention (Line 52) That the snake exhibited “anxiety” as one of its symptoms. That is not an emotion that I have heard associated with a reptile previously.  Perhaps the snake was very active or aggressive, which would be more common descriptors of reptile behavior.
  • The manuscript refers in several places (Lines 52, 94, & 115) to the snake having been fed “commercial pet food.” Captive pythons typically feed on rats or mice, either alive or dead, and certainly such prey can be purchased.  However, it would be unusual to describe such prey simply as “commercial snake food.”  Some snakes can be trained to consume a sausage-like food, but in my experience that is a very uncommon food for captive snakes, and it is unlikely to have been fed to this snake.  However, it is possible that the snake could have been infected from its prey (or commercial food), so that may be an important point to clarify in the manuscript.

The remainder of my comments are stylistic or grammatical, as follows:

Line 3:  “Python regius” should be italicized and “regius” should not be capitalized

Line 13:  “Python regius” should be italicized

Line 15:  Should read “observed in the mouth cavity”

Line 17:  Replaced semicolon with comma

Line 43: “Chryseobacterium” should be italicized

Line 44:  Should read “In the present study:

Line 49:  “Python regius” should be italicized

Line 80:  Should read “Phylogenetic relationships among”

Line 81:  Should read “isolated in the present study”

Line 84:  Should read “strain, in the present study,” and “abscess in the python”

Line 95:  Should read “was disinfected”

Lines 99-100:  Should read “antibiotics are recommended”

Line 105: “indologenes” should not be capitalized

Line 108:  Should read “Ch. Indologenes has previously been”

Line 136-137:  Should read “non-nosocomial isolates showed resistance to fewer antibiotics than we have observed”

Line 141:  Should this read “asymptomatically”, rather than “asymptotically”?

Line 149:  Should read “animal infections should be comprehensively investigated”

Line 150:  Should read “microorganism’s role as an emerging zoonotic”

Author Response

Dear Reviewer 1.

Line 116: We believe that there are many studies, papers and books about oral infections in captive snakes; however, they are not caused by Chryseobacterium and other infectious agents have been implicated in this infection. It is the first time that an oral infection with Chryseobacterium species is reported and we have presented it as a case report in our study. This is mentioned and highlighted in the text.

Line 52: “anxiety” is changed to “aggressive”, revised and highlighted in the text.

Lines 52, 94 and 115: We have found that the animal was fed on preys including rodents and birds. Also, the probable cause of infection because of its prey fed to the animal by the owner is added to the text and highlighted. According to it, it is revised and highlighted in the text.

Line 3: revised and highlighted in the text.

Line 13: revised and highlighted in the text.

Line 15: revised and highlighted in the text.

Line 17: revised and highlighted in the text.

Line 43: revised and highlighted in the text.

Line 44: revised and highlighted in the text.

Line 49: revised and highlighted in the text.

Line 80: revised and highlighted in the text.

Line 81: revised and highlighted in the text.

Line 84: revised and highlighted in the text.

Line 95: revised and highlighted in the text.

Line 99-100: revised and highlighted in the text.

Line 105: revised and highlighted in the text.

Line 108: revised and highlighted in the text.

Line 136-137: revised and highlighted in the text.

Line 141: revised and highlighted in the text.

Line 149: revised and highlighted in the text.

Line 150: revised and highlighted in the text.

Kind regards,

Dr. Pakbin and Dr. Bruck

Reviewer 2 Report

Thank you for this paper. Study design is well-done and the article is well-written.

Clinical and and biological features were well described and of interest. The analysis is reasonable. The results are believable. The discussion and conclusions are supported by the results. I have no ethical concerns.

There is some unhelpful repetition between the introduction and the discussion.

Author Response

Dear Reviewer 2

Unhelpful repetition is removed from the text (in the discussion section).

Other changes are highlighted in the text. 

Kind regards,

Dr. Pakbin and Dr. Bruck